# Feasibility Assessment of a Magnetic Layer Detection Method for Field Applications

**Chenhui Li [1], Liang Liu [2], Mingbin Huang [3,\*] and Yafang Shi [4]**

[1] College of Resources and Environment, Northwest A & F University, Yangling 712100, China; lichenhui@nwafu.edu.cn

[2] Co-Innovation Center for Sustainable Forestry in Southern China, Key Laboratory of Soil and Water Conservation and Ecological Restoration of Jiangsu Province, College of Forestry, Nanjing Forestry University, Nanjing 210037, China; liuliang@njfu.edu.cn

[3] State Key Laboratory of Soil Erosion and Dryland Farming on the Loess Plateau, Institute of Soil and Water Conservation, Northwest A & F University, Yangling 712100, China

[4] School of Horticulture Landscape Architecture, Henan Institute of Science & Technology, Xinxiang 453000, China; shiyf@hist.edu.cn

\* Correspondence: hmbd@nwsuaf.edu.cn

**Abstract:** The lack of current data on the spatial distribution of soil erosion hinders the ability to predict soil erosion on slopes. To address this problem, we present a simple and sensitive soil erosion measurement method called the magnetic layer detection (MLD) method. This method can measure the changes in soil layer thickness (SLT) at a site featuring a pre-buried magnetic layer (ML) using the relationship between magnetic susceptibility and soil thickness, and then use this information to determine the erosion or accumulation at that point. To verify the adaptability and accuracy of this method, we designed three field tests. First, we conducted a simulated sediment siltation experiment using the erosion pin (EP) method as the control. This experiment demonstrated the accuracy of the MLD method for measuring SLT ($R^2_{ad}$ = 0.97, NSE = 88.14%, and RMSE = 1.17 mm). Next, a simulated rainfall experiment with the runoff plot (RP) method as a control was used to demonstrate the effectiveness of the MLD method under water erosion. The results demonstrated that this method can reliably estimate soil erosion on both bare plots ($R^2_{ad}$ = 0.83, NSE = 71.78%, and RMSE = 0.56 mm) and sparse grass plots ($R^2_{ad}$ = 0.90, NSE = 81.77%, and RMSE = 0.25 mm), with performance that is better than that of the traditional EP method. Finally, a soil scouring experiment, again with the RP method as a control, was designed to verify whether the MLD method could accurately measure the erosion of a slope after the soil was scoured by surface runoff. The MLD method could accurately measure the slope erosion ($R^2_{ad}$ = 0.91, NSE = 89.55%, and RMSE = 0.42 mm), with results superior to those of the traditional EP method. The MLD method's results were similar to those from the laser scanner (LS) method, but more accurate and with less associated cost and data processing time. Therefore, the MLD method has potential as a reliable measurement method that can provide useful guidance for elucidating the spatial distribution of soil erosion and predicting slope soil erosion. This approach could be used to obtain raw data to quantify soil erosion on the Loess Plateau of China and beyond.

**Keywords:** magnetic layer detection; erosion pin method; runoff plot method; laser scanner method; sediment siltation experiment

## 1. Introduction

Due to human activities and soil erodibility, the Loess Plateau region has become one of the most eroded areas in the world [1,2]. More erosion monitoring is essential to inform long-term soil conservation in this region [3]; however, obtaining the raw field data of soil erosion in the region remains challenging. Many types of monitoring methods are available, such as erosion pins [4,5], runoff plots [6,7], 3D laser scanners [8,9], real-time

kinematic (RTK)-GPS [10,11], aerial photos or remote sensing [12,13], and tracers [14,15]. Equations for predicting soil erosion have also been developed, such as the universal soil loss equation or revisions thereof [16–18].

However, each method has its associated applicability and limitations. Erosion pins have the potential to disturb the soil and may be covered by a new layer of material or disturbed by both animal and human activities [19]. Additionally, because of their small size and surface exposure, they can be quite challenging to detect, especially in areas with dense vegetation [20]. Runoff plots are typically expensive to maintain, and most relevant studies are based on spatially averaged soil erosion data [21]. Although 3D laser scanners can capture tiny amounts of soil loss [22], the complexity of ground cover in the field severely limits the benefits of this approach. RTK-GPS technology can be used to determine the total amount of soil erosion and its spatial variation by providing accurate elevation information [23]; however, the relatively high cost and poor portability of the instrument limit it from being more widely utilized in the field. Aerial photography or remote sensing can investigate soil erosion on a large scale, but difficulties arise when mapping complex ground conditions with high and dense coverage [24]; in addition, image interpretation is often very labor-intensive [24,25]. Measuring soil erosion using time-domain reflectometry is possible in theory; however, no practical instances of this being done have been reported [26].

Several models have been used to estimate soil erosion in areas with minimal data or under future climate scenarios. However, these models often require high-quality field data [27] and could be significantly improved [28]. An alternative approach is to use tracers to monitor soil erosion. Various types of tracers have provided considerable data on the rates and spatial distribution of soil erosion [29–31]. The main types of erosion tracers include fallout radionuclides (e.g., $^{137}$Cs, $^{210}$Pb, or $^{7}$Be) [32,33], rare-earth elements [34,35], and magnetic materials [36–38]. However, existing tracer techniques also come with limitations. For fallout radionuclides, such as $^{137}$Cs or $^{210}$Pb, unreasonable assumptions of spatial homogeneity, uncertain functional inputs, and radiation risks to researchers and the environment have led to challenges [35,39]. The cost of using rare-earth elements is high, and it takes a tremendous amount of work to prepare, deploy, and analyze the data from such experiments [40–42]. Other techniques have employed stable isotopes, but these require a minimum sample size, and the cost is high [43].

More recent efforts have shifted the focus to magnetic materials as tracers, which can be divided into two types: those related to the existing properties of the soil, and those employing exogenous magnetic materials [21]. The former are based on natural differentiation processes of the topsoil and subsoil to reflect the redistribution pattern of the soil [44,45]. Magnetic susceptibility (MS) is a very important parameter in the identification of soil erosion and is used as a tracer in both types. After the first introduction of MS into soil research by Borgne (1955), related results have been continuously reported [46–52]. However, when there is a large amount of sampling to be done, pretreatment using natural soil magnetism is time-consuming and labor-intensive. Liu et al. (2016) improved the soil core sampling technique and enhanced a core sampler kit, resulting in a method that is more efficient and reliable than the original [53]. Furthermore, several researchers [37,54–56] have chosen to add exogenous magnetic materials directly to the soil to enhance MS. Various types of magnetic powders have been created for this purpose, including magnetic plastic beads [37,38], fly ash and cement [57], a mixture of fine soil, fly ash, cement, bentonite, and magnetic powder [58], and soil passed through a 2 mm nylon sieve with magnetic powder [3,59]. However, these non-soil materials that serve as tracers have different properties than the soil [21], which may affect their utility as erosion tracers. Some efforts have been made to track erosion by heating the soil to enhance the magnetism of the sediment [21,60], but such methods are limited to small areas due to their relatively high cost and the limited types of soils to which they can be applied.

To assess the quantitative sediment redistribution, a new erosion measurement method named the magnetic layer detection (MLD) method was developed by Liu et al. in 2019 [61].

The MLD method determines superficial soil layer thickness by detecting the depth of an underground artificial magnetic layer, using an MS field probe, and then shifts from rapid MS measurement to soil erosion or deposition quantification [61]. Indoor water and wind erosion experiments based on the MLD method have demonstrated advantages such as rapid measurement, relatively low cost, and high accuracy. Liu et al. (2020) evaluated the effectiveness of similar magnetite powder in erodible soils on four soil textures in China [59]. The results indicated that the magnetite powder was suitable for erosion monitoring in China and beyond [59,62]. A MagHut model was developed using magnetic tracer data from Liu et al. (2019) for modeling volumetric magnetizability in both the forward and inverse directions [61,63]. This also implies that this method could also be used to validate process-based erosion models and monitor soil erosion dynamics. However, the effectiveness of the method under natural field environmental conditions in the Loess Plateau region remains to be tested. Despite their shortcomings, some methods of detecting soil erosion, such as the runoff plot (RP), erosion pin (EP), and laser scanner (LS) methods, can return accurate results. Therefore, this study aims to verify the effectiveness of the MLD method by comparing it to the RP, EP, and LS methods under the conditions of simulated sediment siltation experiments and simulated soil erosion experiments in the field.

The objectives of this study were to (1) measure the changes in soil layer thickness (SLT) using the MLD method in simulated sediment siltation, simulated rainfall, and soil scouring experiments, and (2) assess the reliability of the MLD method for measuring soil erosion in the Loess Plateau region by comparing its results with those from the RP, EP, and LS methods.

## 2. Materials and Methods

### 2.1. Study Site

Field studies were conducted in the Wangdonggou watershed (35°12′ N–35°16′ N, 107°40′ E–107°42′ E; total area 8.3 km$^2$), Changwu County, Shaanxi Province, China, in 2019 and 2020 (Figure 1). The study site is dominated by a semi-arid and continental monsoon climate with an annual mean precipitation of 571 mm (with about 70% of annual precipitation falling from June to September), an annual mean temperature of 9.2 °C (1957–2014), and an annual mean open-pan evaporation of 1440 mm (1957–2009) [64,65]. The study site is located in the southern gully area of the Loess Plateau, where the elevation ranges from 946 to 1226 m above mean sea level and the mean slope gradient is 36.4% [64]. According to the FAO-UNESCO soil classification system, the soil texture at the study site is silty clay loam in the 0–10 m profile, with mean sand, silt, and clay contents of 7.7%, 66.2%, and 26.2%, respectively, and corresponding standard deviations of 2.0%, 1.7%, and 3.0%. The porosity in the 0–10 m profile varies from 0.445 to 0.547, with a mean value of 0.498 [66]. The most common plant species in this area include black locust (*Robinia pseudoacacia* L.), Chinese pine (*Pinus tabulaeformis* Carr.), and the perennial grass *Bothriochloa ischaemum* L. The main crops in the study site are winter wheat (*Triticum aestivum* L.) and spring corn (*Zea mays* L.). The growing season for most natural plants is from April to early October.

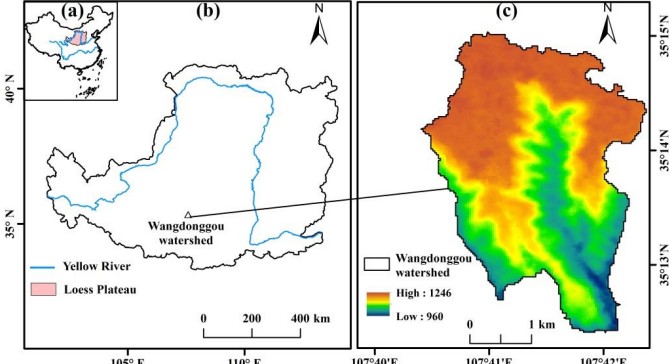

**Figure 1.** Location of the study area: (**a**) China, (**b**) Loess Plateau, and (**c**) Wangdonggou watershed.

*2.2. Measurement Methods*

The three field experiments conducted to evaluate the MLD method are summarized in Figure 2. A detailed comparison of the MLD method and the EP, LS, and RP methods is shown in Table 1.

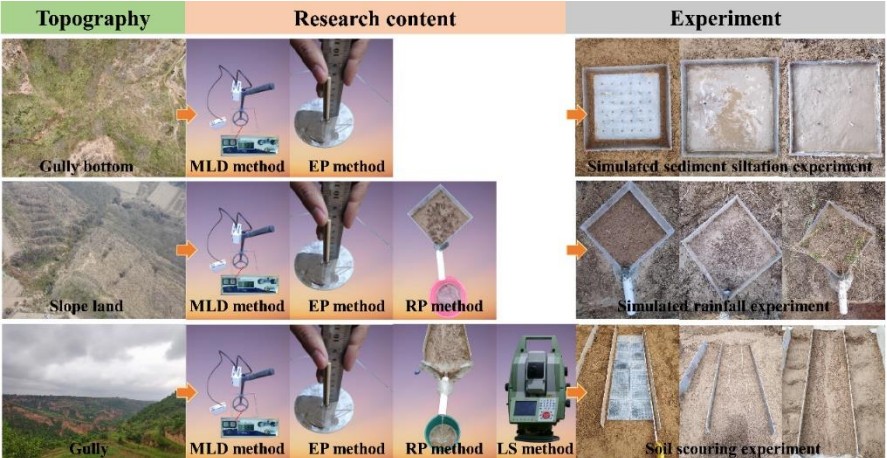

**Figure 2.** Three types of field simulation experiments to assess the MLD method.

**Table 1.** Comparison of four erosion measurement methods: MLD method [61], EP method, LS method, and RP method.

| Item | MLD | EP | LS | RP |
|---|---|---|---|---|
| Working mechanism | Magnetic susceptibility (MS, k) of surface soil layer thickness (SLT, H) is rapidly determined using the MS2D field probe, and then the SLT is calculated by the conversion equation H = f(k); the difference between the two SLTs represents the eroded SLT | The eroded surface SLT is determined using a measuring scale before and after an erosion period | The three-dimensional laser scanner uses laser ranging technology, external reference items, and an internal coordinate system to produce a three-dimensional point cloud | The runoff plot method involves surrounding a specified space, leaving an outlet on one side, and setting up a runoff bucket beneath to collect sediment after a rainfall event |
| Measurement accuracy | <±2 mm within the SLT of 80 mm [61] | Roughly ±1 mm of the eroded surface SLT [67] | 2 ± 0.002 mm [68] | No error by default and used as a control |
| Measurement depth | Roughly 20 cm | Roughly 30 cm | The surface reached by the laser | Depends on the volume of erosion |
| Measurement area | About 572.56 cm² (the MS2D probe covers a circular area with a diameter of 27 cm) | About 70.88 cm² (there is a glass plate with a diameter of 9.5 cm around the erosion pin) | The area of the entire plot | The area of the entire plot |
| Measurement time cost | About 1 s per measurement | About 1 s per measurement | About 0.5 h per measurement | About 9 h per measurement |
| Equipment cost (USD) | About USD 10,000 per instrument (Bartington MS2 meter and MS2D probe) | About USD 1 per stick (measuring scale) | About USD 75,000 per instrument | About USD 100 per set (scouring experiment) |
| Application area | Used in water erosion and wind erosion | Widely used in water erosion and wind erosion | Used in water erosion and wind erosion | Used in water erosion |
| Time required for data processing | About 1 min per treatment | About 1 min per treatment | About 1 h per treatment | About 1 min per treatment |

MLD: magnetic layer detection method; EP: erosion pin method; LS: laser scanner method; RP: runoff plot method.

### 2.2.1. The EP Method

The EP method is a common and widespread approach for monitoring soil erosion [5,69]. This method requires nailing an EP at the site where the soil erosion is monitored. The height of the EP is used as the erosion baseline, and the change in height exposed before and after erosion is used to reflect the erosion deposition conditions. The method is simple in principle, easy to operate, and its monitoring accuracy is good. However, as noted above, important limitations include the potential for soil disturbance caused by the insertion of the pins, along with the possibility that they will be covered by new soil or disturbed by animal or human activity [19]. Moreover, depending on the environment, it may take several rainstorm events or years for substantial erosion to occur [20]. In our study, the EP was passed through a glass plate (9.5 cm diameter, 0.3 cm thick) with an opening of 0.6 cm to achieve accurate measurements (Figure 3d). A steel ruler with a glass plate as the base was used to measure the change in SLT.

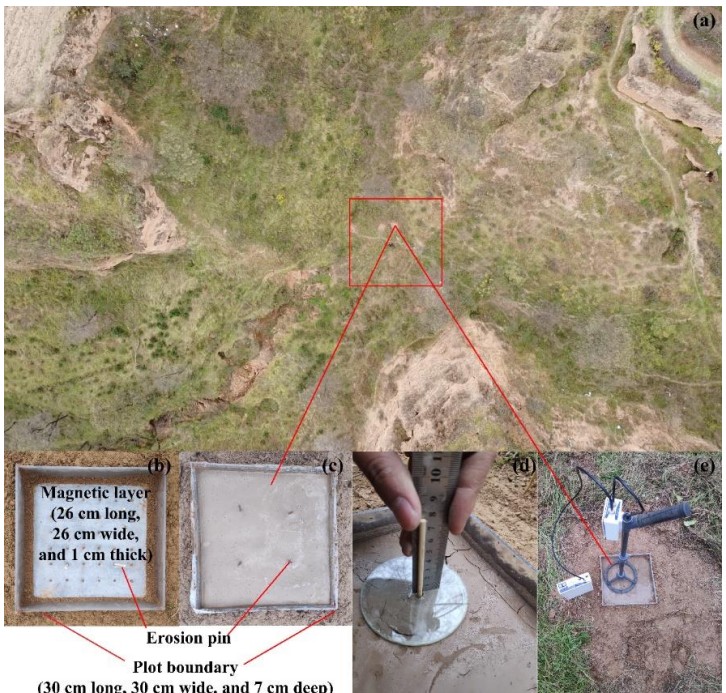

**Figure 3.** Simulated sediment siltation experiment: (**a**) topographical features around the plots; (**b**) plot before the accumulation occurred; (**c**) plot after the accumulation occurred; (**d**) erosion pin (EP) method; (**e**) magnetic layer detection (MLD) method.

### 2.2.2. The RP Method

The RP method is a classic method to measure soil erosion [6,70,71]. The advantage of this method is that the data are accurate and reliable. Runoff plots are frequently damaged during extreme events, which results in the loss of sediment yield events and relevant data [72]. In addition, high maintenance costs and low spatial representation may affect the application of the method [21]. In both the simulated rainfall experiment (Figure 4) and the soil scouring experiment (Figure 5), the RP method served as the control group.

### 2.2.3. The LS Method

Laser scanners can detect minute variations in soil erosion processes and have good accuracy [73,74]. A Leica MS50 laser scanner was used to measure the soil surface profile within the plot before and after each scouring experiment (Figure 5b). Before each scan, the scanning accuracy was set to 1 mm, and three reference points were placed around the plot to ensure each measurement was in the same coordinate system [68].

The scanned data were first preprocessed in Cyclone 6.0.3 and converted to text format, before being imported into ArcGIS (version 10.3, ESRI, USA) for the creation of digital elevation models (DEMs). Thereafter, they were derived with a precision of 1 mm by using the inverse distance weight interpolation method [75] and converted into raster format (*.grid). The difference between the two raster datasets (before and after the scouring experiment) was then calculated using a raster calculator in map algebra. Finally, the average elevation, which represented the variation in soil thickness, was determined. In addition, by extracting the raster data at fixed locations generated from the entire plot, the soil layer variation for each magnetic layer (ML) was identified.

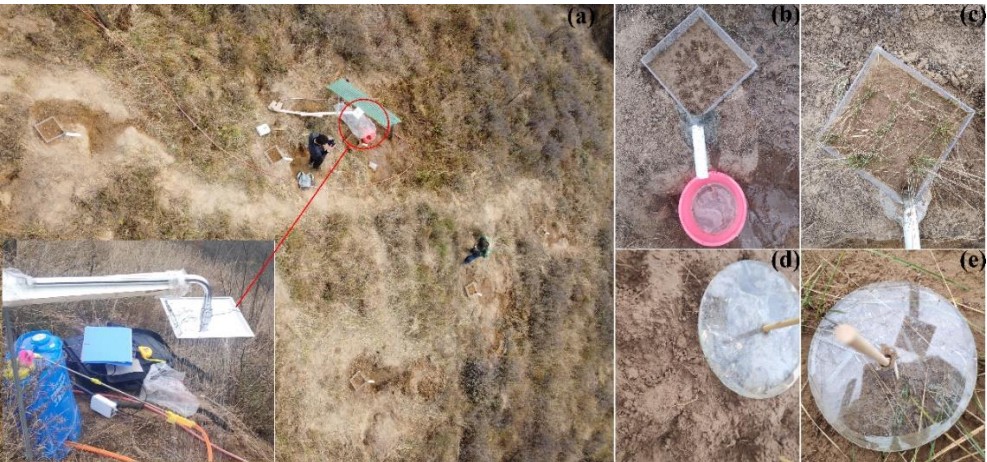

**Figure 4.** Simulated rainfall experiment: (**a**) test scenario and equipment; (**b**) bare soil plot after simulated rainfall; (**c**) grass plot after simulated rainfall; (**d**) details of bare soil plot after simulated rainfall; (**e**) details of grass plot after simulated rainfall.

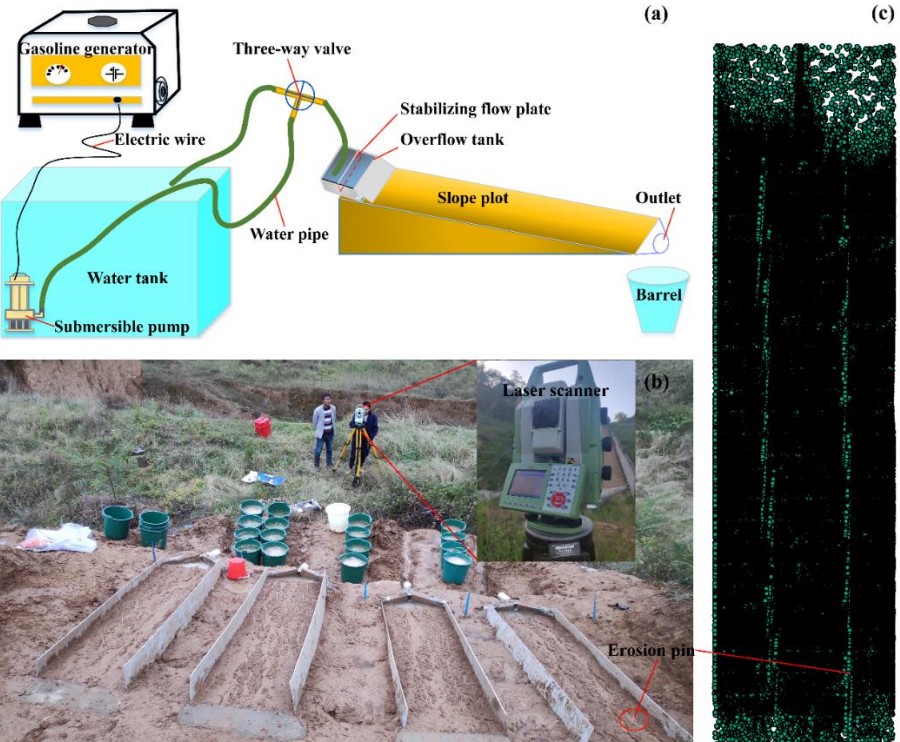

**Figure 5.** Outdoor soil scouring experiment: (**a**) schematic of the experiment; (**b**) laser scanner (LS) method; (**c**) point cloud data of one target runoff plot based on the LS method.

### 2.2.4. The MLD Method

As a new in situ technique, the MLD method can quickly and precisely calculate the SLT by detecting the depth of an underground artificial ML [61]. This enables a shift from rapid MS measurement to soil erosion or deposition quantification. The MLD method is suitable for measuring sheet erosion on the range of slope from 0 to 56%, without the limitation of morphology [61]. The accuracy of the MLD method is considered to express the k-H relationship [61]. The optimal k-H equation is

$$H = 202.68 - 93.46e^{\frac{k}{-0.0067}} - 58.69e^{\frac{k}{-0.0503}} - 54.80e^{\frac{k}{-0.3371}} \tag{1}$$

where k is the value of volume MS, and H is the distance from the ML to the MS2D probe (Bartington Instruments, UK). The k values can be calculated by the following equation:

$$k = \frac{k_0 - k_b}{k_m} \tag{2}$$

where $k_0$ represents the volume MS of the target ML in the MLD method, and k is a deformation of $k_0$. Before and after every sample measurement, blank air readings were taken, and the $k_0$ value was measured five times using the MS2 meter and MS2D probe (Bartington Instruments, UK) [29,76]. The background value ($k_b$) represents the volume MS on the soil surface, while the maximum value ($k_m$) denotes the volume MS on the ML surface.

### 2.3. Field Experiments

### 2.3.1. Simulated Sediment Siltation Experiment

After removing the litter and weeds from the gully bottom, five test plots (30 cm long, 30 cm wide, 7 cm deep) were constructed with acrylic panels (Figure 3). An ML (26 cm long, 26 cm width, 1 cm thick) with four wooden EPs inserted in the fixed position was placed in the center area at the bottom of each plot. The experiment involved the simulated accumulation of sediment above the ML and was repeated five times for each test. To achieve soil layer thickness variation based on sediment siltation greater than 2.6 kg/m$^2$ in the study of Liu et al. (2019), 500 g of soil was sieved through a 2 mm mesh screen and mixed well with 500 g of water before the mixture was poured into the plot [61]. To reduce the error caused by the roughness of the soil surface in the experiment, the mud was spread out as flat as possible (as shown in Figure 3c), and the measurements were taken after the moisture on the soil surface had dried. The accumulated SLT was determined by the MLD method and the EP method. The accuracy of the MLD method was verified using the EP method as the control.

### 2.3.2. Simulated Rainfall Experiment

The simulated rainfall experiment is shown in Figure 4. Three plots of bare soil and three plots of sparse grassland were built on 30° slopes to simulate soil erosion from steep slopes. Four wooden EPs were laid in each plot. A PVC pipe secured with cement was used as the output, which emptied the runoff into a bucket placed below (Figure 4b). Powered by a diesel engine, water was pumped from a storage tank and delivered to the location of the simulated rainfall plots. The rainfall intensity was adjusted by both the return valve and the pressure switch. Each plot (three bare + three grassland) was subjected to four rainfall events, for a total of 24 simulated rainfall events. The simulated rainfall was delivered from a height of 1 m directly above the plots for 1 min. The rainfall intensity was between 30.15 and 113.07 mm/min and was calibrated three times by measuring the flow for 30 s to ensure that the error did not exceed 5%. The average value of the soil bulk density in the test plots was 1.22 g/cm$^3$. For each treatment, the simulated rainfall experiments were repeated three times.

The main vegetation types in the area were *Stipa bungeana*, *Artemisia sacrorum*, and *Agropyron*. Therefore, the grass type used in the test plots was *Stipa bungeana*, which was

transplanted from locations around the plots. The coverage of the grass plots was set at 10, 20, and 30%, respectively. The sediment volumes from the runoff plot method were converted to erosion depths and used to validate the accuracy of the EP method and the MLD method under simulated rainfall conditions in the field. The specific details of the simulated rainfall experiments are shown in Table 2.

**Table 2.** Basic information related to the simulated rainfall experiment.

| No. | Rainfall Events | Treatments | Rainfall Intensity (mm/min) | No. | Rainfall Events | Treatments Coverage (%) | Rainfall Intensity (mm/min) |
|-----|-----|-----|-----|-----|-----|-----|-----|
| 1 | 1 | Bare soil 1 | 67.84 | 13 | 1 | Grassland 1 (10%) | 30.15 |
| 2 | 2 | Bare soil 1 | 90.45 | 14 | 2 | Grassland 1 (10%) | 45.23 |
| 3 | 3 | Bare soil 1 | 105.53 | 15 | 3 | Grassland 1 (10%) | 57.29 |
| 4 | 4 | Bare soil 1 | 113.07 | 16 | 4 | Grassland 1 (10%) | 90.45 |
| 5 | 1 | Bare soil 2 | 90.45 | 17 | 1 | Grassland 2 (20%) | 75.38 |
| 6 | 2 | Bare soil 2 | 90.45 | 18 | 2 | Grassland 2 (20%) | 82.92 |
| 7 | 3 | Bare soil 2 | 97.99 | 19 | 3 | Grassland 2 (20%) | 82.92 |
| 8 | 4 | Bare soil 2 | 105.53 | 20 | 4 | Grassland 2 (20%) | 97.99 |
| 9 | 1 | Bare soil 3 | 37.69 | 21 | 1 | Grassland 3 (30%) | 75.38 |
| 10 | 2 | Bare soil 3 | 75.38 | 22 | 2 | Grassland 3 (30%) | 75.38 |
| 11 | 3 | Bare soil 3 | 90.45 | 23 | 3 | Grassland 3 (30%) | 87.44 |
| 12 | 4 | Bare soil 3 | 105.53 | 24 | 4 | Grassland 3 (30%) | 90.45 |

2.3.3. Soil Scouring Experiment

The experimental system was composed of a water tank, a gasoline generator, a submersible pump, some water pipes, a three-way valve, an overflow tank with a stabilizing flow plate, some barrels, and slope plots (Figure 5). The water used for the experiment was supplied from a water pond at the bottom of Wangdonggou watershed, which was pumped into a water tank (5 m long, 2.5 m wide, and 2 m deep) near the plots. The test flows were controlled by adjusting the switch of a three-way valve. Three flow discharge rates (3, 6, and 9 L/min) were employed based on local rainfall information from the study site [77] and the size of the plots. The flow discharge rates were calibrated three times by measuring the flow for 1 min to ensure that the error did not exceed 5%. Each experiment lasted for 10 min, which was timed with a stopwatch. Slopes of 20° and 25° were used in the experiment, which are slightly less than the average slope of 36.4% determined from a field survey [64]. Two replicates of the two slope gradients were conducted (Figure 5). The size of the ML (26 cm long, 26 cm wide, and 1 cm thick) was taken into consideration in the plot design (2 m long, 0.55 m wide, and 0.1 m deep). On the surface of the plots, the ML was organized into seven rows and two columns, with a wooden EP inserted in a fixed position (Figure 5b). After pressing and leveling, the ML was covered by a soil layer in the test plots. The plots were refilled and flattened before each experiment, with a 24 h interval between experiments on the same plot.

Before and after the experiment, the soil bulk density of the plots was determined, and it ranged from 1.25 to 1.27 g/cm$^3$. The sediment samples collected by the runoff bucket were dried and weighed to calculate the soil erosion amounts. Values were calculated using the MLD, EP, and LS methods.

### 2.4. Data Analysis

OriginPro 2023 (OriginLab Inc., Northampton, MA, USA) and Microsoft Office 2013 (Microsoft Inc., Washington, DC, USA) were used to plot all figures. The performance of the MLD, EP, and LS methods was evaluated using the adjusted coefficient of determination ($R^2_{ad}$), root-mean-square error (RMSE), and Nash–Sutcliffe efficiency (NSE) [78]. The coefficient is commonly used to assess model performance, with values greater than 70% indicating good performance, values between 40% and 70% indicating satisfactory performance, and values below 40% indicating unsatisfactory performance [79].

## 3. Results and Discussion

### 3.1. Assessing the MLD Method Based on the Simulated Sediment Siltation Experiment

Figure 6 depicts the SLT measured by the MLD and EP methods (*n* = 50). The results show a good linear relationship ($R^2_{ad}$ = 0.97). The RMSE based on all data is 1.17 mm, which falls within the theoretical accuracy range of the MLD method (<2 mm) [61]. The NSE (88.14%) also demonstrates that the equation is a good fit to the data. In general, the values resulting from the MLD method are higher than those from the EP method. The difference between the two sets of results increases at higher SLT values.

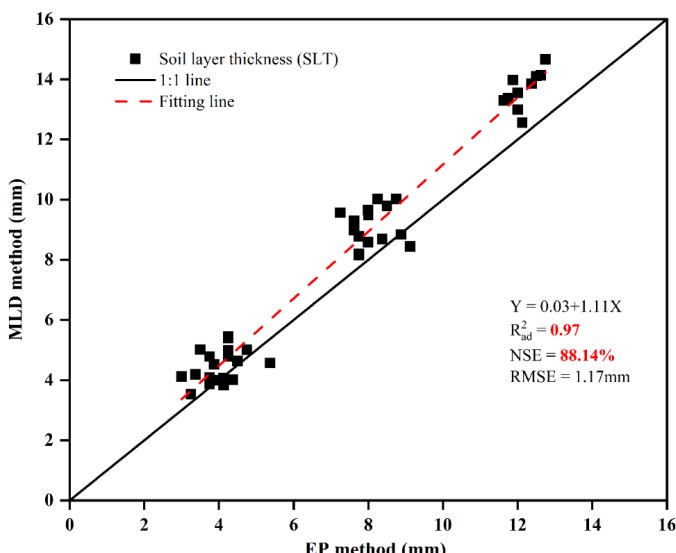

**Figure 6.** Comparison between measured accumulated SLT based on the EP method and the MLD method in the simulated sediment siltation experiment. $R^2_{ad}$: adjusted R-squared; NSE: Nash–Sutcliffe efficiency; RMSE: root-mean-square error; EP: erosion pin; MLD: magnetic layer detection.

The EP method has been widely used for quantifying soil erosion and accumulation [5,80,81], and it was used as a control in this experiment. The flattening of the soil layer surface following each simulated sediment accumulation might be the reason for the good agreement of the results for the two methods. In addition, most of the area of the plot was measured multiple times, providing numerous data points that also helped with the fitting of the results. Due to the EP being short and close to the soil surface, the operator was often in an overhead position when reading the data; this might have resulted in smaller values from the EP method and indirectly led to higher values for the MLD method, as well as the trend that became more obvious as the thickness of the soil layer increased. In addition, the EPs were manually placed and their tops were not perfectly parallel to the surface of the plots, which was not exactly horizontal after the simulated sediment accumulations. As such, reading with the scale on the steel ruler could have led to errors in the EP method that would not occur in the MLD method; this could be an additional reason for the consistently higher values recorded using the MLD vs. the EP method.

In summary, the accuracy of the SLT measured by the MLD method is in the order of millimeters and reliable compared to the EP method. In other words, for a given SLT measurement (<13 mm, based on the data herein), the values determined by the MLD method are a satisfactory representation of the results obtained by the EP method in the simulated sediment siltation experiment.

### 3.2. Assessing the MLD Method Based on the Simulated Rainfall Experiment

SLT measurements based on the MLD method were compared with those based on the EP method on bare soil plots in the simulated rainfall experiment (Figure 7a). The NSE of data from the EP method vs. the RP method (control) was $-59.43\%$, indicating that the accuracy was poor. The adjusted coefficient of determination ($R^2_{ad} = 0.28$) also supports this view. In contrast, data from the MLD method showed a better fit with the RP method (NSE = 71.78% and $R^2_{ad} = 0.83$). Furthermore, the RMSE of SLT data for the RP vs. MLD methods was 0.56 mm, which reflects a better fit than for the RP vs. EP methods (1.34 mm).

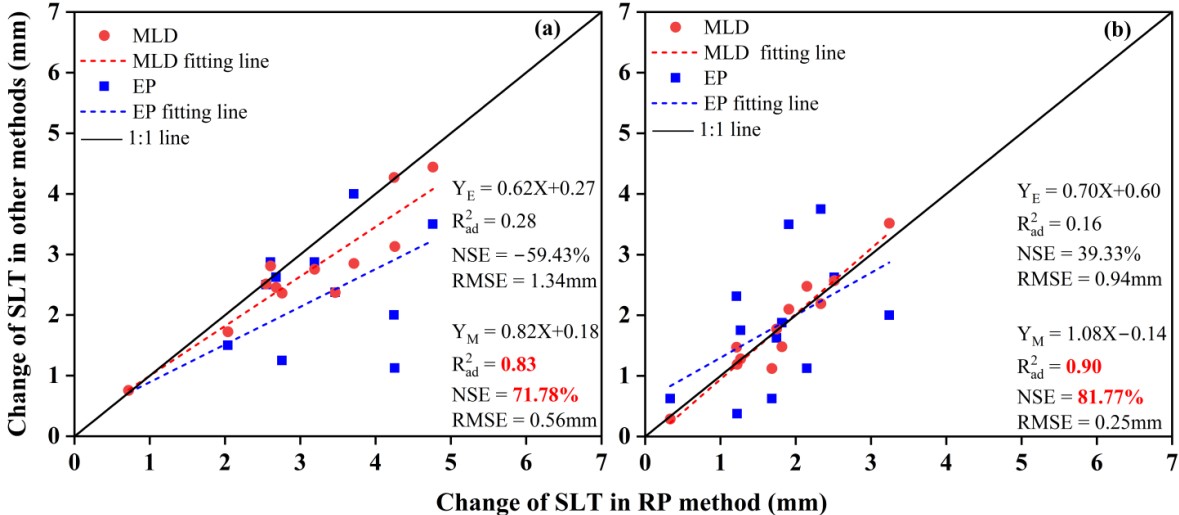

**Figure 7.** Changes in SLT in the simulated rainfall experiment, as measured by the RP method vs. the EP and MLD methods in (**a**) bare soil plots and (**b**) grass plots. $Y_E$: SLT measured by the EP method; $Y_M$: SLT measured by the MLD method; $R^2_{ad}$: adjusted R-squared; NSE: Nash–Sutcliffe efficiency; RMSE: root-mean-square error; RP: runoff plot method; SLT: soil layer thickness; MLD: magnetic layer detection method; EP: erosion pin method.

The reason for the poor performance of the EP method and the good performance of the MLD method, compared to the RP method (control), is likely due to various aspects of these measurement methods. The ground surface was not homogeneously erosive after rainfall, which resulted in the EP method not being able to measure erosion under the glass plate due to poor spatial representativeness (Figure 4b,d). This is assumed to be the main reason why the EP method underestimated the variation in soil thickness compared to the RP method (Figure 7a). In the MLD method, the probe (circle with a diameter of 27 cm) occupied almost the entire plot (square with a side length of 30 cm), with the value measured representing the mean of the area that it covered. Liu et al. (2019) concluded that the method works better for sheet erosion brought on by wind or water agents [61]. For such uneven surfaces (Figure 4b,d), the MLD method is unable to determine minor changes in erosion depth within the probe diameter and, thus, also underestimates the erosion measured by the RP method. Overall, the MLD method underestimates values measured using the RP method, but to a lesser extent than the EP method (Figure 7a).

Again using the RP method as a control, Figure 7b depicts the trend of eroded soil thickness measured by the EP and MLD methods in the grass plots. There is a poor fit between the data collected using the EP method and the RP method ($R^2_{ad} = 0.16$ and NSE = 39.33%). In contrast, the data collected using the MLD method (*n* = 12) demonstrate

good agreement (NSE = 81.77%) with the RP method within the eroded depth of 4 mm in the grass plots, as well as an acceptable linear relationship ($R^2_{ad}$ = 0.90). In addition, the associated RMSE (0.25 mm) is close to the theoretical accuracy of the MLD method (<2 mm), suggesting that the theoretical accuracy of the MLD method for eroded SLT is satisfactory in the grass plots in the simulated rainfall experiments.

The change in SLT was smaller in the grass plots than in the bare soil plots, which might have been due to the presence of vegetation cover reducing soil erosion [82,83]. However, the random distribution of grass increased the surface roughness of the measurement points in the plots, making it difficult for the EP measurement disk to fit well to the surface. As a result, the measurement accuracy of the EP method was reduced [61]. The grass in the plots also posed a challenge with respect to the placement of the MS2D probe in the MLD method, but this did not seem to affect the measurement results (Figure 7b). After simulating the rainfall, the surface of the grass plots was relatively flat compared to the bare soil plots, which might account for the better results obtained by the MLD method.

### 3.3. Assessing the MLD Method Based on the Soil Scouring Experiment

The SLT measurements based on the RP method were compared with those from the MLD, EP, and LS methods (Figure 8). Plotting data measured using the RP method vs. data from the MLD and LS methods showed a good linear relationship between the SLT values ($R^2_{ad}$ = 0.91 and 0.94, respectively). The estimated NSE was 89.55% and 90.62% for the MLD and LS methods, respectively, showing that they were successful in reflecting the change in SLT measured by the RP method. The RMSE of 0.42 mm for the RP vs. MLD methods falls within the theoretical accuracy range of the MLD method (<2 mm). In general terms, under the change in SLT considered (0.1–3.6 mm), the LS method overestimates while the MLD method underestimates values determined using the RP method. This trend increases as the change in SLT increases. Compared to the other two methods, the EP method has the worst fit ($R^2_{ad}$ = 0.52, NSE = 81.77%, and RMSE = 1.22 mm) and tends to underestimate values determined using the RP method.

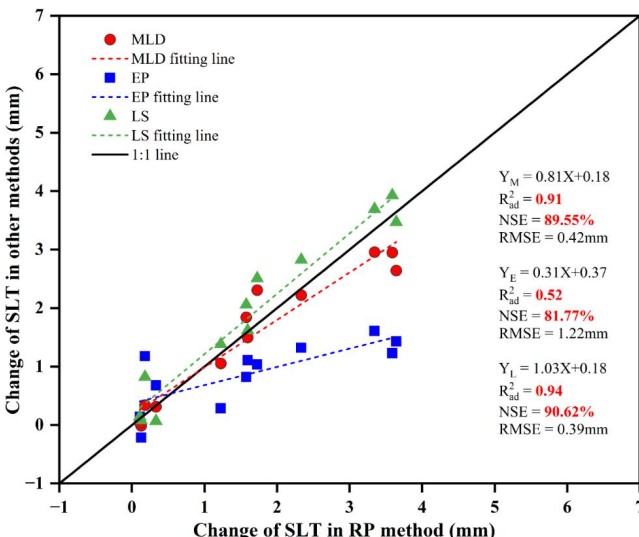

**Figure 8.** Changes in SLT in the soil scouring experiment measured by the RP method vs. the EP, MLD, and LS methods. $Y_E$: SLT measured by the EP method; $Y_M$: SLT measured by the MLD method; $Y_L$: SLT measured by the LS method; $R^2_{ad}$: adjusted R-squared; NSE: Nash–Sutcliffe efficiency; RMSE: root-mean-square error; RP: runoff plot method; SLT: soil layer thickness; MLD: magnetic layer detection method; EP: erosion pin method; LS: laser scanner method.

In this experiment, the EPs blocked runoff and caused sediment to accumulate around them (Figure 5b), which might have resulted in large measurement errors [84]. This explains the poor fit between values measured using the EP and RP methods (Figure 8). Images based on point cloud data that are similar to the ML size must be cropped when the LS

method is used to determine the change in SLT on each ML. This can easily lead to large measurement errors and, thus, overestimate values compared to the RP and MLD methods. Thus, data processing based on the LS method is highly dependent on operator skills. In the MLD method, however, no surrounding materials should affect the measurement of MS. In this experiment, the ML was connected and fixed in position, therefore generating small measurement errors. The spacing of the ML could be widened in future experiments to avoid this problem.

Fiener et al. (2018) used various techniques (tracers, terrestrial laser scanning, and unmanned aerial systems) to quantify erosion [54]. The results revealed that all techniques had individual error sources, and erosion measurements made under almost optimal conditions were still subject to significant uncertainty. Daniel et al. (2019) evaluated and compared three methods (EP, RP, and LS) for measuring erosion [85]. It was found that the three methods have limited comparability. While the LS method can collect clear, high-resolution data on streambanks, the EP and RP methods are more accurate representations of complicated vegetated streambanks. This may explain the large variation in results between different methods in our study.

Taking the RP method as the control, all data obtained using the MLD method are shown in Figure 9 for the various experimental conditions. Most of the measured values from the MLD method vary within the range of 2 mm when the average SLT ranges from 0.1 to 3.6 mm. The data measured using the MLD method fluctuate slightly at a flow rate of 3 L/min (Figure 9a,b), with this fluctuation tending to increase at flow rates of 6 L/min (Figure 9c,d) and 9 L/min (Figure 9e,f). No obvious difference was noted between data for the two slopes tested (20° and 25°).

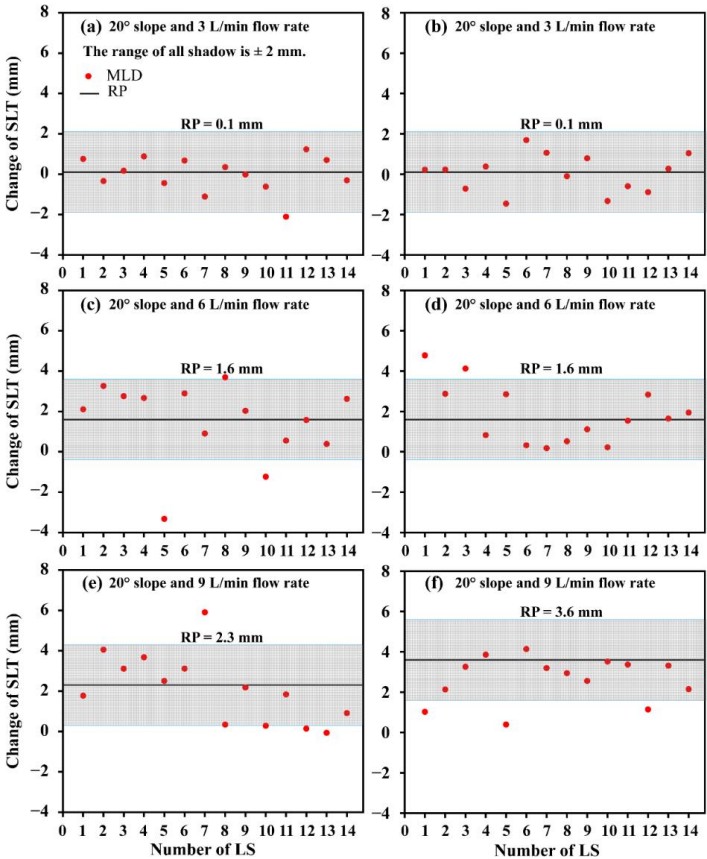

**Figure 9.** The difference in SLT between the MLD method and the RP method under different treatments in the soil scouring experiment; (**a**–**f**) represent the combined treatment of slope and flow rate in the experiment. ML: magnetic layer; SLT: soil layer thickness; MLD: magnetic layer detection method; RP: runoff plot method.

Most of the data from the MLD method varied within 2 mm under all conditions, consistent with the precision of its millimeter-level measurement. The increase in flow rates resulted in more erosion, which caused the surface to become more undulating. If the surface roughness is sufficiently great to affect the probe's contact with the soil surface, the error of the MLD method might rise [61]. Few data points varied by more than 2 mm, which might have been due to the MLD data representing the average SLT within an area of 572 cm$^2$. However, the overall low variability demonstrates the stability and accuracy of the method. The similar findings from the two slopes (20° and 25°) can be attributed to the small difference in slope.

## 4. Conclusions

The MLD method could be a new method for soil erosion measurement. The effectiveness of the MLD method was tested in the field in a natural environment by comparing it with three common soil erosion measurement methods (RP, EP, and LS). The method was able to accurately measure the changes in soil layer thickness when soil accumulation occurred in the simulated sediment siltation experiment ($R^2_{ad}$ = 0.97, NSE = 88.14%, and RMSE = 1.17 mm). Using the RP method as the control, the method also performed sufficiently well in the simulated rainfall experiment ($R^2_{ad}$ = 0.83, NSE = 71.78%, and RMSE = 0.56 mm), and it was even able to resist the interference of sparse grasslands ($R^2_{ad}$ = 0.90, NSE = 81.77%, and RMSE = 0.25 mm). In the soil scouring experiment, the MLD method effectively measured the changes in SLT on the steep slopes (20° and 25°) ($R^2_{ad}$ = 0.91, NSE = 89.55%, and RMSE = 0.42 mm) at the plot scale when compared to the traditional RP method. The measurement error of the MLD method was generally less than 2 mm at the point scale.

The MLD method has limitations related to perturbations in the topsoil when the ML is first laid, as well as the millimeter scale of its measurement accuracy. However, it was demonstrated herein to be a low-cost, fast, and relatively high-precision soil erosion measurement method (Table 1) that is not affected by vegetation occlusion and can efficiently obtain raw data of onsite erosion. This method thus represents an important option for dynamic monitoring and modeling of slope erosion. In the future, it would be interesting to study different types and materials of artificial ML for reducing perturbations in the topsoil when the ML is first laid. In general, the MLD method has good potential for application in the Loess Plateau region and beyond.

**Author Contributions:** C.L.: investigation, methodology, writing—original draft, writing—review and editing, C.L., L.L. and Y.S.: formal analysis. L.L. and M.H.: supervision, resources, writing—review and editing. All authors have read and agreed to the published version of the manuscript.

**Funding:** This work was supported by the National Natural Science Foundation of China (Grant Nos. 42277352 and 41701310) and the Key Science and Technology Research and Development Program of Henan Province, China (Grant No 232102321042).

**Institutional Review Board Statement:** Not applicable.

**Informed Consent Statement:** Not applicable.

**Data Availability Statement:** Publicly available sources of the data used in this study are described in the article. However, other relevant data or information can be requested from the authors on reasonable grounds.

**Acknowledgments:** All authors are deeply grateful to the editors and anonymous referees for their valuable suggestions to improve the quality of this paper.

**Conflicts of Interest:** All authors declare that they have no conflict of interest.

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
