# Peer review of "Feasibility Assessment of a Magnetic Layer Detection Method for Field Applications"

_sustainability, doi:10.3390/su151914263_

Round 1
Reviewer 1 Report
- The researchers addressed the feasibility of using magnetic layer detection method as an erosion measurement method, in terms of stability and accuracy.
- After I compared the research question (1) with others in this field of study, I can say the topic is original, relevant and addresses a specific gap in the field (i.e. absence of original data on the spatial distribution of soil erosion).
- In filling this specific gap, researchers provided original data on the spatial distribution of soil erosion for predicting soil erosion on slopes.
- As regards methodology, and as a way of further study, authors are advised to verify the feasibility of data capture using this method on significantly varying slopes as opposed to the small difference in slope (20° and 25°) that was considered for this study. As a way of control, the authors already acknowledge synergy of methods for more reliable results.
- Yes, the conclusions are consistent with the evidence and arguments presented and they do address the main question posed?
- Yes, the references are appropriate.
- The tables and figures are well designed and well represented in the main text.
Reviewer 2 Report
1- Line 37, I think this approach can be used in other region of the world with loess plateau such as Iran, see "Identification of Expansion Rate in Active Gullies using Remote Sensing". You can revise this statement and appropriate following section to extend this research application.
2- Line 63, what do you mean by "complex ground cover"? did you mean the high and dense forest?
3- Fig1, section c, does the polygon represent the watershed area as legend said? While in the left region it doesn't tie to higher elevation I think it may represent other land feature?
4- Lines 143-208 focus on method description. Can such describing text can present as a new table?
5- Line 209, does the loess moisture also test and analyzed?
6- While I was reading table 2, This question came to me, what is the natural condition of the vegetation in the investigated area in different months of the year? What types of crops are cultivated in the loess cover of this area and how is the monthly distribution of rainfall in this area?
7- The chosen topic for the article needs to be modified so that it can better introduce the research work done.
8- Providing more complete explanations regarding the time of tests and their costs can be used by researchers to expand the use of this method
9- How was the degree of porosity, granularity and homogeneity of the soil in the samples investigated and what percentage of importance of these items could be in the results?
The text was fluent
Reviewer 3 Report
Feasibility assessment of a magnetic layer detection method for field applications based on three common erosion measurement methods
1. In the introduction, the authors present long previous methods that have been applied for erosion measurement, however, none of the reference citations mention accuracy or the previous studies. Please, provide the accuracy of the previous studies to compare your accuracy in the discussion section.
2. In the method, only part of methods presents the replication of the measurements.
3. Line 125, there is a misunderstanding of soil type. The authors do not mention the soil type, but only mention the properties of the soil. The examples of soil type are: Vertisols, Inceptisols, and soon.
4. Table 2, the simulation used extremely high rainfall intensity, normally intensity is in mm/hour, not mm/minute, rainfall intensity 97mm/minute is extremely high, I am not sure whether it can be happened in the really environment. With annual rainfall 571 mm, the applied rainfall intensity 97mm/minute is unrealistic.
5. The result for MLD method has R2 extremely high 0.97, is it due to flattening the soil layer surface following each simulation of sediment accumulation, so it does not represent the natural environment.
6. Line 325 – 330, that is the reason why the EP method has lower accuracy than that of MLD method, so if it cannot be compared these methods, because there is a disturbance in the EP method.
7. Line 424, the number of measurements for MLD is only 12, it is not enough to draw a scientific conclusion.
8. The slopes of the equations in Figure 7 are low, although there is a certain high R2 values. The low slope of an equation meant the change in independent variable is not really change the value of the dependent variable

Reviewer 4 Report
1. In page 3 before the section of material and method
the type of erosion which can be predicted with the m
ethod presented in the paper should be described.
2. Is method capable in different slopes and morphology
Page 3 paragraph 1 line 1 quantity the restirbution
of sediments change to the quantity of sediments restribution
Reviewer 5 Report
Hello
Thank you much for the opportunity given to me to review the attached manuscript. After a careful reading of the work hinging on the assessment of soil erosion rate using Magnetic Layer Detection (MLD) in loess region in China, in comparison with other common methods, I found it an interesting subject but immature in presentation. The manuscript was started well and finished unsuitable. The work from the viewpoint of the subject and goal was potentially a good one. But, some exaggeration has been made to highlight the merits of the MLD techniques. In real conditions, no one can be substituted by each other. It gives some information that would be interesting to the journal's readership but there was no rationale for the work.
- The present gap in the study field has not been highlighted specifically to verify the necessity and novelty of the present work. It has to be properly justified why this treatments have been taken into consideration.
- No review of literature and state-of-the-art of the existing problems in the study field with further focuses on international and recent papers in the same field has been made.
- The entire research methodology needs to be well-justified and documented. It deemed to me that it has been fully planned arbitrarily.
- The entire results and discussion would be reviewed after getting assured about the methodology and study procedure!!
- The practical use of the research findings has to be highlighted at the end of the manuscript in the conclusion. The translation of research results to practical purposes is a vital task for the researcher to be deeply and explicitly considered. I am not sure that such a type of results with potential issue dealing with lab experiments can be welcome by the users and policymakers.
- Conclusion is not also concise and research specific. It has not provided the necessary information for the readership to lead them to proper and adaptive studies in the future. It has to contain a concise summary of the key findings of the study. Furthermore, it is not necessary to summarize the study or to discuss the methodology. Further, general managerial implications and future research can be briefly pointed out in the conclusion.
Overall, despite the valuable attempts of the authors, it suffers some flaws and also serious challenges and deficits as annotated as 50 comments in the reviewed manuscript and briefly mentioned above which makes me against its acceptance for publication in the present form. However, major substantial revision for further review and after insuring incorporation of all comments and suggestion is recommended.
Best Wishes

Round 2
Reviewer 2 Report
The limitations mentioned in the respond letter should be included in the manuscript text , the rest of the amendments is approved
Minor corrections is needed in manuscript text
Author Response
1. The limitations mentioned in the respond letter should be included in the manuscript text , the rest of the amendments is approved
Response: The following sentence has been added in the text to show the limitations.
“The MLD method has limitations related to perturbations in the topsoil when the ML is first laid”
2. Minor corrections is needed in manuscript text
Response: We have checked the manuscript again and revised some minor errors.
Reviewer 3 Report
Dear authors,
I appreciate for your paying attention to the suggestions from the previous review. Please improve the introduction as I indicate using green color.
Regards

Author Response
The comment shown in the PDF file:
"What I meant in my first review was what is the R2 from Liu's paper, as Liu presented in Table 3 or Figure 7, the quantitative value of the accuracy, and it is not just like in Table 1 in this paper."
Response: After checking Liu's paper (Liu et al. 2020), we could not find the R2 values and other related values. No R2 values can be shown in our manuscript.
Reviewer 5 Report
Dear Editor
Hello
Considering revisions made in the manuscript, it can be then accepted for publication. However, some comments like rain erosion substitution, were not addressed.
Best Wishes
Dear Editor
Hello
Considering revisions made in the manuscript, it can be then accepted for publication. However, some comments like rain erosion substitution, were not addressed.
Best Wishes
Author Response
Considering revisions made in the manuscript, it can be then accepted for publication. However, some comments like rain erosion substitution, were not addressed
Response: We have used “water erosion” instead of “rain erosion” in the new version.